# Carbon and Nitrogen Isotope Ratios of Food and Beverage in Brazil

**DOI:** 10.3390/molecules25061457

**Published:** 2020-03-24

**Authors:** Luiz A. Martinelli, Gabriela B. Nardoto, Maria A. Z. Perez, Geraldo Arruda Junior, Fabiana C. Fracassi, Juliana G. G. Oliveira, Isadora S. Ottani, Sarah H. Lima, Edmar A. Mazzi, Taciana F. Gomes, Amin Soltangheisi, Adibe L. Abdalla Filho, Eduardo Mariano, Fabio J. V. Costa, Paulo J. Duarte-Neto, Marcelo Z. Moreira, Plinio B. Camargo

**Affiliations:** 1Laboratory of Isotope Ecology, Center for Nuclear Energy in Agriculture, University of São Paulo, Av. Centenário, 303, São Dimas, Piracicaba CEP 13416-000, SP, Brazil; mazperez@cena.usp.br (M.A.Z.P.); garruda@cena.usp.br (G.A.J.); fffracass@cena.usp.br (F.C.F.); jgiovann@cena.usp.br (J.G.G.O.); isadoraottani@hotmail.com (I.S.O.); sarahlima0723@gmail.com (S.H.L.); eamazzi@cena.usp.br (E.A.M.); tgomes@cena.usp.br (T.F.G.); soltangheise@gmail.com (A.S.); adibefilho@cena.usp.br (A.L.A.F.); dumariano@gmail.com (E.M.); mmoreira@cena.usp.br (M.Z.M.); pbcamargo@cena.usp.br (P.B.C.); 2Ecology Department, Institute of Biological Sciences, University of Brasília, Asa Norte, Brasília CEP 70910-900, Brazil; gbnardoto@gmail.com; 3National Institute of Criminalistics, Federal Police, Asa Sul, Brasília CEP 70610-200, Brazil; mr.f.bio@gmail.com; 4Department of Statistics and Informatics, Rural Federal University of Pernambuco, R. Manuel de Medeiros, 35, Dois Irmãos, Recife CEP 52171-050, Brazil; pjduarteneto@gmail.com

**Keywords:** processed foods, staple foods, photosynthesis metabolism, isotopes, Brazil

## Abstract

Several previous studies on targeted food items using carbon and nitrogen stable isotope ratios in Brazil have revealed that many of the items investigated are adulterated; mislabeled or even fraud. Here, we present the first Brazilian isotopic baseline assessment that can be used not only in future forensic cases involving food authenticity, but also in human forensic anthropology studies. The δ^13^C and δ^15^N were determined in 1245 food items and 374 beverages; most of them made in Brazil. The average δ^13^C and δ^15^N of C_3_ plants were −26.7 ± 1.5‰, and 3.9 ± 3.9‰, respectively, while the average δ^13^C and δ^15^N of C_4_ plants were −11.5 ± 0.8‰ and 4.6 ± 2.6‰, respectively. The δ^13^C and δ^15^N of plant-based processed foods were −21.8 ± 4.8‰ and 3.9 ± 2.7‰, respectively. The average δ^13^C and δ^15^N of meat, including beef, poultry, pork and lamb were -16.6 ± 4.7‰, and 5.2 ± 2.6‰, respectively, while the δ^13^C and δ^15^N of animal-based processed foods were −17.9 ± 3.3‰ and 3.3 ± 3.5‰, respectively. The average δ^13^C of beverages, including beer and wine was −22.5 ± 3.1‰. We verified that C-C_4_ constitutes a large proportion of fresh meat, dairy products, as well as animal and plant-based processed foods. The reasons behind this high proportion will be addressed in this study.

## 1. Introduction

Rapid industrialization and urbanization together with changes in lifestyles are among the factors responsible for changes in eating habits, especially the consumption of highly processed foods [1]. The access, mainly by urban populations, to an immense variety of processed food products triggered the so-called “supermarket era” [2]. Diversification as a marketing strategy has resulted in a vast range of food products in the last two decades. However, consumers may be duped into buying food items without being aware of their quality and integrity [3].

Brazil is one of the world’s main food consumers which makes the country a vast market for food fraud and adulteration through the large quantity of foods produced, exported, or imported [4]. In the Brazilian territory, complex supply chains of foods with animal origin, such as milk and dairy products, are the main targets of food fraud and adulterations, followed by vegetable oils, especially olive oil, considered as high-value products. Meat and fish, as well as their respective by-products, were also involved in some food fraud and adulteration [4]. Recently there has been a couple of scandals refocusing national attention to food adulteration. The “*Ouro Branco*” and “*Carne Fraca*” operations conducted by the Brazilian Federal Police revealed the addition of illegal substances in milk and beef, respectively, to increase business profits. The operations received public attention due to the health harm risks that those adulterations inflicted.

Since sophisticated fraudsters have been generally capable of mimicking the physical and chemical compositions of targeted foods, there is a need to address and prevent food fraud and adulteration [4]. Food safety might be also compromised in these cases. In this context, a current tool of choice in food forensics is stable isotope analysis, which can be a rapid and cost-effective method to detect fraud [3]. Several studies on targeted food items using carbon and nitrogen stable isotope ratios in Brazil revealed that many of the food items investigated are adulterated, mislabeled or even fraudulent [5,6,7,8]. In this regard, carbon isotope ratio analyses have become a useful tool to track the abundances of C_3_ and C_4_ food sources in human and animal diets [9]. The primary photosynthetic producers support hundreds of millions of heterotrophic species like humans. The C_4_ photosynthetic pathway evolved in plants 24–35 million years ago as a response to the decrease in CO_2_ and increase in O_2_ atmospheric concentrations [10]. By the competitive advantage of these plants in warm environments, extensive grasslands were formed, mainly in low latitudes [11].

Several large terrestrial herbivores, like bovine, equid and other ungulates, followed by top carnivores, and, finally, hominids, have adapted to these new ecosystems [12]. The presence of hominids in grasslands was remarkable as such in human evolution [10]. The importance of C_4_ plants became even more evident when humans started agriculture, since in this endeavor, C_4_ plants like maize (*Zea mays* L.), sorghum (*Sorghum bicolor* L.), pearl millet (*Pennisetum glaucum* L.) and sugarcane (*Saccharum* spp.) were among the species selected by humans to be cultivated [13]. Maize had such an impact in the Americas that in few centuries, the human populations changed from a C_3_ to a C_4_-based diet [14]. Furthermore, with the Columbian exchanges, C_4_ plants were disseminated throughout the old and new continents [15,16,17].

Even with this competitive advantage, there is still a large dominance of C_3_ species, almost four times more than C_4_ species. The European Union (EU) and other temperate regions preferentially grow C_3_ rather than C_4_ plants, which modifies their food carbon isotopic composition. As an example, Martinelli et al. [18] observed that hamburgers in countries that feed cattle preferentially with C_4_ grass or maize (e.g., Brazil, Mexico, and USA) have less negative *δ*^13^C values in comparison to countries with cattle preferentially fed with C_3_ plants (e.g., the EU).

Unlike carbon, of which the main source for terrestrial plants is atmospheric CO_2_, nitrogen can be acquired from both the atmosphere through biological fixation and through the soil. After the Haber–Bosch process became a viable method at industrial scale, commercial crops also commenced to take up nitrogen from mineral fertilizers. Before the large-scale use of these fertilizers, crops had relied mainly on manures, organic sources of nitrogen generated by animals. Due to the multiple nitrogen sources for plant uptake, interpreting the nitrogen isotopic ratio (^15^N/^14^N) in plants is not an easy task [19,20]. However, the use of fertilizers with δ^15^N values around 0‰ can be distinguished from the use of manures with more positive δ^15^N values [21,22]. Thus, as a first approach, δ^15^N of crops may be an indicative of the application of synthetic or organic fertilizers for improving plant growth (e.g., [23]).

After being incorporated into the plant tissue, nitrogen moves up along the food web with preferential loss of ^14^N related to ^15^N along the way. Consequently, the organisms positioned higher in the food chain tend to be more enriched in ^15^N than the organisms positioned on the bottom of the chain, the so-called “trophic isotopic discrimination” [24]. Therefore, the nitrogen isotope ratio has been a useful tool to place organisms along the food chain, including humans. Vegetarians and vegans tend to be depleted in ^15^N in comparison with omnivorous humans [25].

Carbon and nitrogen isotopic ratios also have the potential to serve as tracers of dietary habits and migration in humans and animals [26,27]. Consequently, the isotopic analysis of human remains such as hair, nails, bones or teeth, allows the retrieval of information regarding a person’s diet to reconstruct dietary habits (e.g., [28,29]) or to determine various regions where the decedents have been fed through time, which gives evidences of past mobility patterns [30].

Previous studies performed in Brazil have found that large urban populations have a relatively homogenous diet regardless of the geographic region of the country [25,31]. Based on the stable isotope composition of fingernails, these authors also found that plants with a C_4_ photosynthetic metabolism (sugarcane, maize and pastures) constitute the major part of the diet in these large urban centers. Based on this fact, we hypothesize here that there is a pervasive presence of C-C_4_ in Brazilian food items although the staple foods in Brazil are rice (*Oriza sativa* L.), beans (*Phaseolus* spp. and *Vigna* sp.), and cassava (*Manihot esculenta* L.), plants that follow the C_3_ photosynthetic metabolism.

In this context, we developed the first Brazilian baseline assessment using carbon and nitrogen stable isotope ratios of about one thousand food items considered essential to test the above hypothesis. We also generated forensic data supporting food authenticity and traceability. As the assessment included in natura as well as processed foods, it might also become a powerful baseline for tracking human movement under the “supermarket diet” trend in Brazil.

## 2. Results

The complete dataset containing the δ^13^C and δ^15^N values of every single sample of foods and beverages is available in the Appendix A.

### 2.1. Plants Used as Food

Considering all samples of in natura plants, there were two main distinct groups observed: those following the C_3_ photosynthesis metabolism and those using the C_4_ pathway (Figure 1a). There were also some plants which follow the crassulacean acid metabolism (CAM), such as pineapple (*Ananas comosus* L.), and their δ^13^C is intermediate between C_3_ and C_4_ plants (Figure 1a). The average δ^13^C of C_3_ plants was −26.7‰ (*n* = 434), varying from −30.7 to −22.5‰, while the average δ^13^C of C_4_ plants was −11.5‰ (*n* = 47), varying from −14.2 to −10.2‰ (Figure 1a), which is a highly significant difference (*p* < 0.01). The values in the CAM plants varied from −15.3 to −14.1‰, with an average of −14.8‰ (*n* = 3; Figure 1a). The δ^15^N of in natura plants had a large variation from −4.4‰ up to more than 18.4‰ (Figure 1b). However, most of them had positive values. The average for C_3_ plants was 3.9‰, which does not differ from the δ^15^N of C_4_ plants (4.5‰; Figure 1b).

Then, according to the FAO classification, plants were grouped into C_3_-cereals, C_4_-cereals, fruits, pulses, tubers and vegetables. Except for C_4_-cereals, the average δ^13^C values of these groups ranged from −30.7 to −14.1‰ (Figure 2a). The main δ^13^C difference was between C_4_-cereals and the other plant groups (*p* < 0.01). On the other hand, δ^15^N values of oilseeds (−0.3‰) and pulses (2.4‰) were significantly less positive (*p* < 0.01) in comparison with other the groups. In addition, nuts (17.3‰) had more positive δ^15^N values (*p* < 0.01) than vegetables, tubers, fruits, and C_3_-cereals, whose average δ^15^N values varied between −4.4‰ and 15.9‰ (Figure 2b).

### 2.2. Meats

Beef had the least negative δ^13^C values (*p* < 0.01) among different types of meat, close to the δ^13^C of C_4_ plants. The δ^13^C of lamb, pork, and poultry were also more negative than that of beef (*p* < 0.01) (Figure 3a). On the other hand, freshwater wild fish had the most negative δ^13^C values (*p* < 0.01). In contrast, δ^13^C values in farmed-raised tilapia (*Oreochromis niloticus*) were similar to those values of pork and poultry (Figure 3a). In addition, marine fish had a similar δ^13^C, but it was more negative (*p* < 0.01) than pork and poultry due to the reasons which will be discussed later.

The δ^15^N of animal protein consumed in southeast Brazil varied considerably among different products (Figure 3b). The most positive δ^15^N values were observed in wild freshwater and marine fish (*p* < 0.01); however, δ^15^N of farmed raised tilapia was only more positive (*p* < 0.01) than pork and poultry (Figure 3b).

### 2.3. Processed Food

The isotope values of processed foods are summarized in the tables due to the large number of food items (Table 1 and Table 2). Plant-based processed foods have a wide range of δ^13^C values (Table 1). Food items with C_3_-like values are flours from wheat (*Triticum* spp.) and cassava, noodles and pasta. Alongside these products, cocoa (*Theobroma cacao* L.) powder and bitter chocolate had also δ^13^C values resembling C_3_ plants, as well as vegetal fat (Table 1). Conversely, sugarcane-derived sugar, fruit, jams, juice-powders, chocolate-drink powders, and industrial puddings had typical δ^13^C values of C_4_ plants (Table 1). Other products are clearly a mixture of C_3_ and C_4_ plants in different proportions. For instance, chocolate bars (bitter, milk, and white) have different δ^13^C values according to the contents of cocoa, milk and sugar. Chocolate powder for cooking, with an average δ^13^C of −19‰, represents a mixture of chocolate and C-C_4_ sugar.

The δ^15^N values of plant-based processed foods varied from 2‰ to 6‰ in most of the cases (Table 1). Products derived from wheat like flour, noodles, pasta, and cookies had δ^15^N values resembling those of in natura wheat grain (see Appendix A). The exceptions are seasonings and stocks that had more negative δ^15^N (*p* < 0.01), which differed greatly from their feedstock, suggesting a high degree of isotopic discrimination in these products (Table 1).

Among processed foods of animal origin, dairy products followed the δ^13^C values of milk, however, some dairy products, such as heavy cream and ice cream, tended to have more negative δ^13^C values compared to milk, although this difference was not significant (Table 2). Processed meat products, like hot dogs, sausages, salami, ham, and lard followed the δ^13^C of pork meat (Table 2). The same pattern was observed for jello made mainly of bovine collagen (Table 2). Dehydrated meat stock cubes had an average δ^13^C value of −19.5‰ (Table 2). By looking at the ingredients of these cubes on their labels, it is difficult to discern whether these δ^13^C values resemble the presence of animal fat tending to have more negative δ^13^C values in comparison with the meat itself or if they resemble the addition of vegetable oils of C_3_ plants with also more negative δ^13^C values. The same conclusion is applicable to meat-based soup powders, sauces, and seasonings with δ^13^C values close to the stock cubes (Table 2).

The δ^15^N values of dairy, processed meat, and jello were similar to those values for milk, pork and beef, respectively, implying that there was a small isotopic discrimination during processing (Table 2). On the other hand, meat-based seasonings, primarily dehydrated meat stock cubes, meat-based seasoning had a significantly less positive δ^15^N value (*p* < 0.01) related to other types of meat (Table 2), therefore suggesting a strong isotopic discrimination during processing and/or a significant amount of plant- rather than meat-derived ingredients (Table 1).

As a classical method of data presentation in stable isotope studies, δ^13^C-δ^15^N biplot (δ-space) is a bidimensional space, giving information about resources consumed by an organism and also the bioclimatic context in each such organism developed, the so-called “isotopic niche” as defined by Newsome et al. [32]. There is a clear separation in the δ-space between in natura food plants and most animal proteins (Figure 4a). Processed foods of animal origin resemble raw meat and poultry in the δ-space (Figure 4b), while processed foods of plant origin seems to have C_4_ and C_3_ plants in variable proportions as ingredients (Figure 4c).

### 2.4. Beverages

The average δ^13^C of wine samples was −23.1‰ *(n* = 291) with a wide range of values, from a minimum of −28.4‰ to a maximum of −14.8‰ (Figure 5a). The average δ^13^C of beer samples was −20.9‰ (*n* = 78), having a bimodal frequency distribution, with values concentrating around −27‰ and around −19‰ (Figure 5a). A few samples of soda were also analyzed just to confirm the heavy presence of sugar. The average δ^13^C of five soda samples was −11.7‰, with a standard deviation of only 0.5‰ (Figure 5b).

## 3. Discussion

The δ^13^C of C_3_ and C_4_ plants used as food were in the range expected considering other studies with native plants [33,34,35,36,37]. Overall, crops are cultivated in a way to avoid shadow and maximize light exposition [38]. Conversely, leaves in a forest must struggle to maximize their exposition to sun light. Consequently, crops also have high evaporative demands, which prompts stomata closing to avoid water losses [36]. As photosynthesis pumps CO_2_ out from the stomata, *pi/pa* (i.e., intercellular to ambient CO_2_ partial pressure) tend to decrease, leading to an increase in the plant δ^13^C of crops in relation to forest leaves. This difference is obvious in Figure 6a, showing a frequency distribution of δ^13^C values of C_3_ crops in this study in relation to δ^13^C of tree leaves of the Brazilian Atlantic forest (Martinelli et al. under review). There is not much overlap between δ^13^C of crops and forest besides a difference of ~6‰ between them (Figure 6a).

There was large variability in the δ^15^N of domesticated plants, from −4‰ to 18‰ (Figure 6b). This trend was expected since plants take up nitrogen from several sources depending on its availability and the climate, and each source has a different nitrogen isotopic ratio [19]. In domesticated plants, in addition to natural sources, soil amendments, such as mineral and organic nitrogen fertilizers, also have to be considered [21]. The primary source of reactive nitrogen before the advent of the Haber–Bosch process (which allowed the production of synthetic fertilizers) was atmospheric nitrogen which becomes available for organisms through biological fixation [39]. The δ^15^N of the nitrogen fixed is ~0‰ and as mineral nitrogen fertilizers are also synthesized by using atmospheric nitrogen, the δ^15^N of such fertilizers is also close to 0‰ [22]. Therefore, plants capable of nitrogen fixation or amended with mineral nitrogen fertilizers tend to have less positive δ^15^N values compared to other plants. This explains why some domesticated plants like pulses (nitrogen-fixing plants of the Fabaceae family) have less positive δ^15^N values compared to other domesticated plants in this study [40].

Huelsemann et al. [41], based on a large survey of the literature, showed that the average δ^15^N of vegetables under the influence of mineral fertilizers was 3.1‰, increasing to 6.8‰ and 9.8‰ in vegetables fertilized with organic and animal-derived manure, respectively. The average δ^15^N of vegetables found here was 4.2‰, which is more positive than the value found by Huelsemann et al. [41] for plants that received mineral fertilizers (Table 3), but less positive than those fertilized with organic manure. The reasons for this difference are difficult to pinpoint, mainly because several nitrogen sources available for domesticated plants. However, it is likely that the following hypotheses can explain this divergence: (i) the more positive δ^15^N values of tropical soils relative to temperate soils [42]; and (ii) a higher amount of nitrogen fertilizer is used in more developed countries than in Brazil [25].

Bovine, poultry and processed pork meat, together with dairy products and eggs, are the most important sources of animal proteins for Brazilians [43]. Most cattle herds in Brazil are free-range and C_4_ grass-fed, especially those of the genus *Brachiaria*. The average δ^13^C of 51 samples of these grasses collected in the Amazon region was −12‰, and the average δ^15^N was equal to 2.0‰ (Martinelli, unpublished data). Poultry and pork in Brazil are raised intensively and fed by a feed composed of maize and soy (*Glycine max* (L.) Merrill) that have the following isotopic composition: δ^13^C = −16.7‰, and δ^15^N = 0.6‰ [9]. Consequently, Brazilian beef, poultry and pork (Figure 4a), as well as dairy products such as milk, heavy cream, yogurt and cheese, and also all types of processed meat products like ham, salami, sausages, hot dogs, among others, are rich in C-C_4_ (Figure 4b). Although lamb is not widely consumed in Brazil, the C-C_4_ is also present in this meat, but less than beef, poultry and pork (Figure 4a). In comparison with other countries, the McDonald’s hamburger made in Brazil had the largest amount of C-C_4_ [18]. The same pattern was found by Osorio et al. [44], who analyzed dry defatted meat from Europe and US. Even in comparison with Chinese defatted beef, which is raised on maize and C_3_ plants, C-C_4_ in Brazilian beef is likely higher than Chinese beef [45].

Most of the forage species used to feed dairy cattle in Europe are C_3_ plants and the addition of maize in their diet is being considered as a deviation from the traditional milk production [46]. Therefore, milk produced in Europe also tends to have more negative δ^13^C values compared to Brazilian milk, although a direct comparison is not possible because the milk fat was not removed in our study [47].

C_4_-C is also found in Argentinian milk, but not at the same proportions which exists in Brazilian milk, because in addition to maize, C_3_ plants like soy, cotton (*Gossypium hirsutum* L.) seeds, alfalfa (*Medicago sativa* L.) and oats (*Avena sativa* L.) are also used to feed dairy cattle in the former country [48]. As milk is the basic dairy ingredient, all dairy products in Brazil tend to have a higher C_4_ content in comparison with dairy products from other countries. The same is true for processed pork meats like ham, which, in Brazil, has a larger proportion of C-C_4_ than dry cured ham raised in Spain [49] and Italy [50]. A similar pattern was observed for Brazilian lamb relative to lamb raised in Europe or in some parts of South Africa [51,52].

The presence of C-C_4_ in animal proteins was even observed in farm-raised fish (Figure 4a), especially tilapia, where the production is growing in the country and is widely available in grocery stores [53]. The fish food which is being used to raise farm-raised fish species is also based on maize and soy, and consequently, the average δ^13^C of farm-raised tilapia is −18.2‰ (Figure 4a). Marine fish also have δ^13^C values resembling animals fed with C_4_ plants (Figure 3a), and the reason is the fact that marine phytoplankton, which are the base of oceanic food chains, have average δ^13^C values close to −21‰ [54], resulting in δ^13^C values of oceanic food chains close to the values resembling a mixture of C_3_ and C_4_-C in terrestrial ecosystems. In contrast, in freshwater systems, the average δ^13^C of the phytoplankton in Brazil is more negative than −30‰ [55], which results in the most negative δ^13^C in freshwater wild fish in comparison with all types of meat (Figure 4a).

Wild fish, either from freshwater or from the ocean, had the most positive δ^15^N values among all types of animal-derived foods (Figure 4a). This is explained by the fact that the food chain in rivers and in the ocean is complex, including several trophic degrees, which increases the isotopic trophic discrimination regarding nitrogen and, therefore, the ^15^N enrichment of the food [56]. δ^15^N in farm-raised fish like tilapia was less positive than wild fish since the available food chain of a farm system is based on fish foods produced from maize and soy (Figure 4a). Coletta et al. [9] also observed that the δ^15^N of free-range chicken was more positive than barn-raised chicken, probably because free-range chicken feed on soil invertebrates. This trend was also observed here between wild animals (bushmeat) and domesticated ones since bushmeat had a more positive δ^15^N (*p* < 0.05) than any domesticated animal (Figure 4). We hypothesize that the animal protein produced from large-scale operations tends to have lower δ^15^N than wild-animal protein since agriculture is a simplification of natural ecosystems (which have more a complex food chain [57]), therefore leading to a low “trophic discrimination”. In addition, soy, a nitrogen-fixing plant that has a low δ^15^N value, is extensively used as animal feed [58].

Among plant-based processed foods, excluding those that are wheat-based like pasta, noodles, crackers, and snacks, C-C_4_ is present in variable amounts in several products, including those for which C-C_4_ occurrence is somehow unexpected (Figure 4c). For instance, commercial fruit jam and preserves have much more C-C_4_ related to C-C_3_ in their compositions. C-C_4_ can also be found in large quantities in soup powders, juice powders, sugary beverages, baby formulas, and spices like cumin (*Cuminum cyminum* L.), saffron (*Crocus sativus* L.), and garam masala (an India-style blend of ground spices), which are mixed with a fine maize flour (termed in Brazil as *fubá*) (Figure 4b). Maize is also present in large quantities in beers made by the largest brewer companies in Brazil (Figure 5). Most Brazilian wines have at least 25% C-C_4_ due to the use of sugarcane as an adjunct in the grape (*Vitis vinifera*) fermentation process [6]. Other products also have suspiciously high contents of C-C_4_, among them, sauces like mustard and ketchups. Perhaps the most iconic example of the heavy presence of C-C_4_ in Brazilian processed foods is soy sauce (*shoyu*), which has a dominant proportion of maize rather than soy or wheat (both are traditional ingredients of the classic recipe) [7].

A question arises about the reasons for such pervasive use of C-C_4_ by the Brazilian food industry. We mention here that the cost of C-C_4_ in Brazil is overall cheaper than C-C_3_. Brazil is mostly a tropical country where the growth of C_4_ plants is favored by a high light incidence [11,12,59]. Therefore, there is a climatic control in the geographical distribution of C_4_ plants and it seems that this distribution can be tracked by certain types of foods. This is the case of hamburgers made by a global fast-food company. The country-level δ^13^C of patties grouped by latitude clearly showed that there was an inverse correlation between the proportion of C-C_4_ and latitude, meaning that countries in the tropical belt had more C-C_4_ than countries located in higher latitudes [18].

In Brazil, among agricultural commodities, two C_4_ plants are very important: sugarcane and maize. Brazil is one of the largest producers of sugar from sugarcane in the world. Brazil was the top sugar producer in 2014, producing ~37 million Mg of sugar [60]. Brazil is also an important producer of maize in the global food market. In 2017, Brazil was the third largest producer of this crop after the USA and China, producing almost 100 million Mg [61]. Therefore, sugar in Brazil is a relatively cheap product which is being widely used by the population and by the food industry. In some remote rural communities of the country, such as those in the Amazon region, sugar is used as a source of energy in some periods of the year during which food is in short supply [62].

The price of maize grain in Brazil is lower than rice, soy, and wheat [58] and consequently, it is widely used to feed animals in Brazil as well as in processed foods. For instance, in 2017, Brazil was the second largest producer of chicken meat (almost 40 million Mg), and beef (almost 10 million Mg); and the fifth largest producer of pig meat, with almost 4 million Mg of meat. Chickens and pigs in the country are fed a combining ratio of maize and soy, with maize as the dominant proportion [9] and most Brazilian cattle herds are fed with C_4_ forage grasses, mainly of the genus *Brachiaria*. Therefore, although raw meat is a relatively expensive item for Brazilians, any meat included in their diets would be a source of C-C_4_ (Figure 4a,b). On the other hand, processed meat, especially pork sausage, termed *calabresa* in Brazil, has become a cheap alternative for the low-income population, primarily in areas where refrigeration is not available, such as isolated communities of the Amazon region [62]. Coupled with the abundance of cheap C-C_4_ in the country, there is also the fact that the Brazilian food legislation is usually very flexible regarding the use of ingredients, allowing the use of maize or sugarcane (sources of C-C_4_) instead of C_3_-derived plants in products such as wine [6] and soy sauce [7].

By investigating a significant number of food items, we believe that it is fair to conclude that although the traditional staple foods of Brazilians are composed mainly of plants that follow the photosynthetic C_3_ metabolism (e.g., rice, beans, and cassava), C-C_4_ plants have a higher degree of pervasiveness in the Brazilian food system (Figure 4), leading to high unaware consumption of these plants by Brazilians (especially processed food [63]), which is ultimately reflected in the more positive δ^13^C tissue values found in the national population, confirming our initial hypothesis.

Finally, addressing and preventing food fraud and adulteration requires not only enforcement of regulatory systems but also more sampling, monitoring, and development of cost-effective methods contributing to fraud detection [4]. In this sense, understanding and interpreting carbon and nitrogen isotope ratios of both plant and animal based food items together with the use of the raw isotope database would support actions and investigations regarding food adulteration and might assist the prevention of food fraud in the Brazilian territory.

## 4. Methods

### 4.1. Sampling Protocol

We analyzed the stable carbon and nitrogen abundances in 1619 samples (804 were composed of in natura vegetal and animal products, while 815 samples were processed foods). Of the total, 1245 and 374 were food items and beverages, respectively. Most of the samples were bought during 2015–2019 in grocery stores located in Piracicaba, a municipality in the Southeast Region of Brazil with a population of ~400,000 inhabitants. In natura animal samples include the following species: cattle, chicken, pig, lamb, freshwater and marine fish, and bushmeat (feral pig and peccary). In natura vegetal samples include cereals, fruits, pulses, tubers, and vegetables grouped according to the FAO classification. For processed food, the classification proposed by Monteiro et al. [64] was followed. Processed foods with animal origin included dairy, cured meat, meat stock, and lard. Dairy products included yogurt, milk cream, ice cream, cheese, and butter. Cured meat included sausages, ham, hot dogs, salami, mortadella, and canned cook pork. Meat stock is defined here as industrially reduced meat stock made from beef, poultry, and pork (Figure 1). Plant-based processed food included: flour, oil and fat, sweeties, chocolates, sauces and soups. “Flour” included the following plants: cassava, maize and soy. “Fats” included margarine and coconut oil. “Sweeties” included a diverse variety of products: corn (maize) flakes, cereal bars, cookies, cakes, pastries, jams, and puddings. “Chocolate” included chocolate bars and chocolate powders mixed with milk (similar to powders used to prepare hot chocolate). “Sauces and soups” included: dehydrated soups, salad sauces, mustard, ketchup, soy sauce (*shoyu*), and Worcestershire sauce. Finally, in the group defined as “Others”, the following items were included: powder spices, like cumim, urucum, saffron, and sweeties like marshmallow, Japanese soy-based products like miso paste, instant misoshiro powder, baby formula, and pre-cooked polenta.

### 4.2. Isotopic Analysis

Solid foods were oven-dried at 60 °C to a constant weight. Subsequently, samples were finely ground to facilitate homogenization. Processed foods were composed of several ingredients and cookies with chocolate chips and cookies stuffed with cream were ground together. After homogenization, 1–2 mg of sample was transferred to a tin capsule for further elemental and isotopic analysis. Beverages like soda, beer, and wine were directly placed in tin capsules.

The isotope ratios of carbon (^13^C/^12^C) and nitrogen (^15^N/^14^N) of each sample were determined using a continuous-flow isotope ratio mass spectrometer (Delta Plus, ThermoFisher Scientific, Bremen, Germany) coupled to an elemental analyzer (CHN-1110, Carlo Erba, Rodano, Italy) at the Laboratory of Isotope Ecology of the Center for Nuclear Energy in Agriculture, University of São Paulo.

Carbon and nitrogen isotope compositions were calculated as
$$\delta \;\left(‰\right)=[\left(R_{sample}/R_{standard})-1\right]\times 1000$$
where *R* is the ratio of ^13^C/^12^C or ^15^N/^14^N. Stable isotope ratios were measured using an internationally recognized standard and relative to a laboratory standard. Thus, we used the 25-(Bis(5-tert-butyl-2-benzo-oxazol-2-yl) thiophene (BBOT; C_26_H_26_N_2_O_2_S; Fisons Instruments Inc., Danvers, MA, USA) as an international standard while fine-milled sugarcane leaves were used as a laboratory standard.

### 4.3. Statistical Analysis

Descriptive statistics (mean, standard deviation, median, inter quartile range, and minimum and maximum values) were used to report the δ^13^C and δ^15^N values of food and beverage samples. To test the differences between food items, ANOVA was used because carbon and nitrogen isotope values followed a normal distribution. The post-hoc Tukey test was used to pinpoint specific differences between food items. Statistical analyses were performed in R (version 3.6.3, The R Foundation for Statistical Computing, Boston, MA, USA) and RStudio (version 1.2.5019, RStudio, Vienna, Austria) using the “multcomp” package [65].

## Figures and Tables

**Figure 1 molecules-25-01457-f001:**
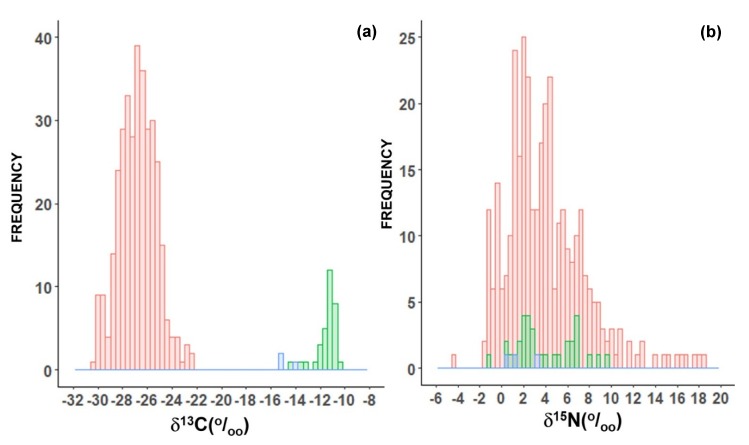
Frequency histogram of δ^13^C (**a**) and δ^15^N (**b**) *of in natura* plants used as food in Brazil. The green bars represent C_4_ plants, the blue bars represent CAM plants, and the red bars represent C_3_ plants.

**Figure 2 molecules-25-01457-f002:**
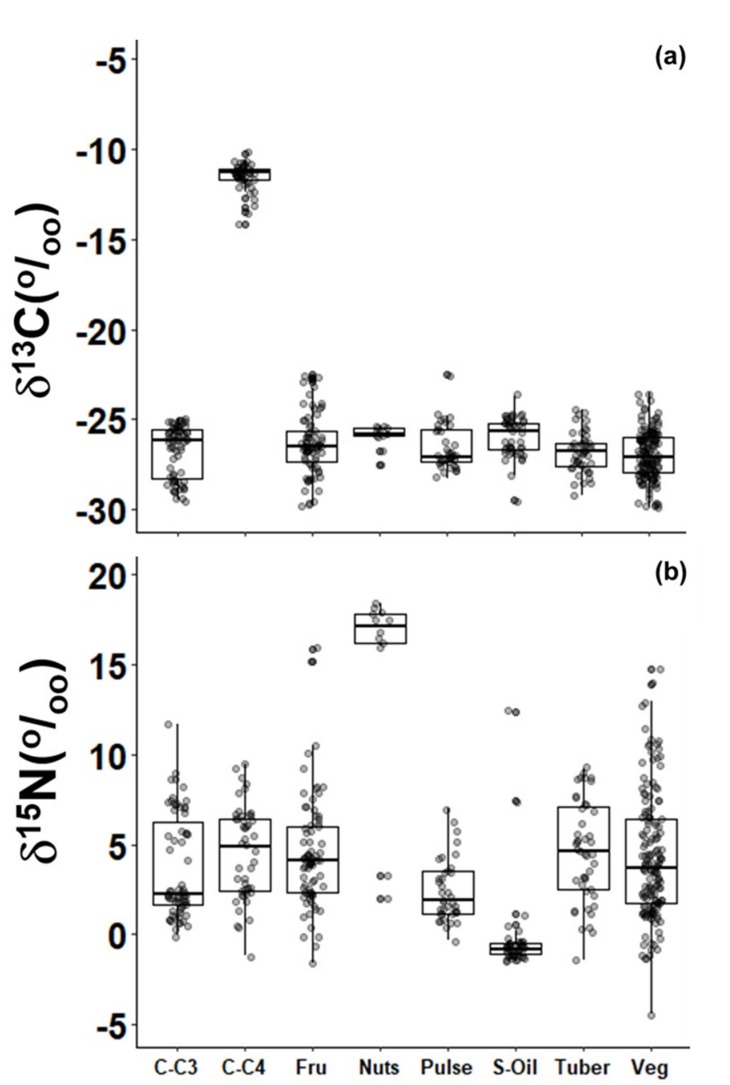
Box-whisker of δ^13^C (**a**) and δ^15^N (**b**) of in natura plants used as food in Brazil grouped according to the FAO classification. C-C_3_: cereals that follow the C_3_ photosynthetic metabolism (wheat and rice); C-C_4_: cereals that follow the C_4_ photosynthetic pathway (maize); Fru: fruits; Nuts: nuts; Pulse: pulses; S-Oil: oilseeds; Tuber: tubers; Veg: vegetables. See the Appendix A for food items included in each of these groups.

**Figure 3 molecules-25-01457-f003:**
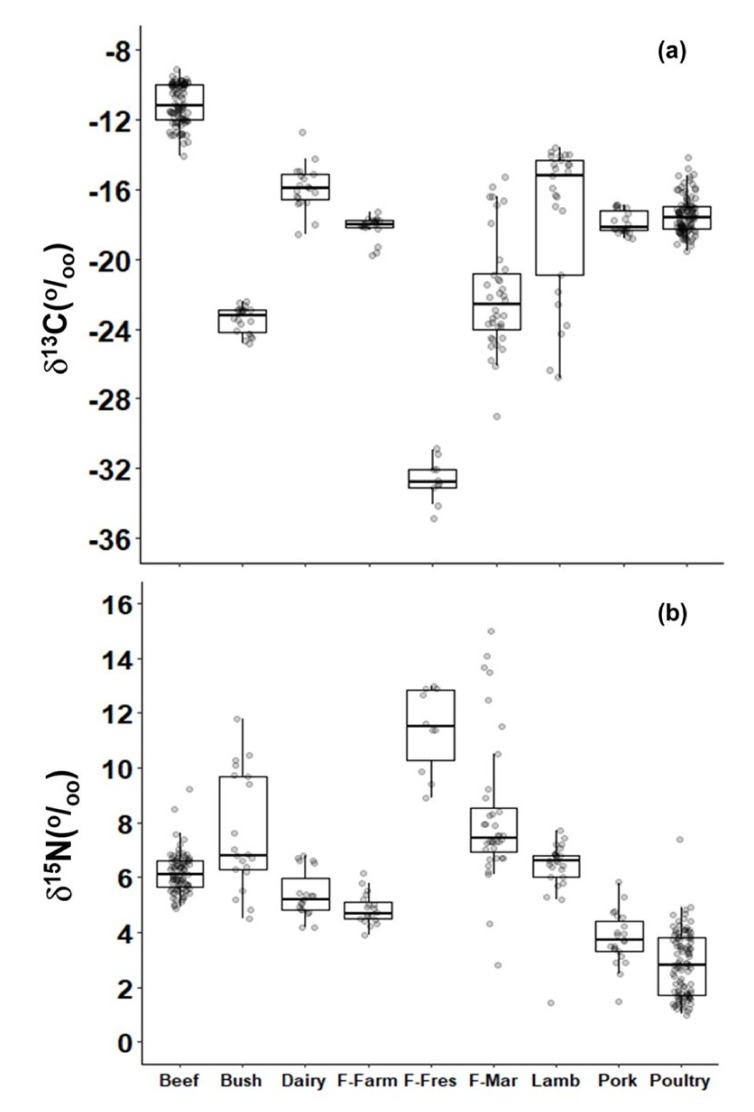
Box-whisker of δ^13^C (**a**) and δ^15^N (**b**) of raw meats used as food in Brazil grouped as follows: beef, bushmeat (Bush), farm-raised fish (F-Farm), freshwater fish (F-Fres), marine fish (F-Mar), dairy, lamb, pork, and poultry.

**Figure 4 molecules-25-01457-f004:**
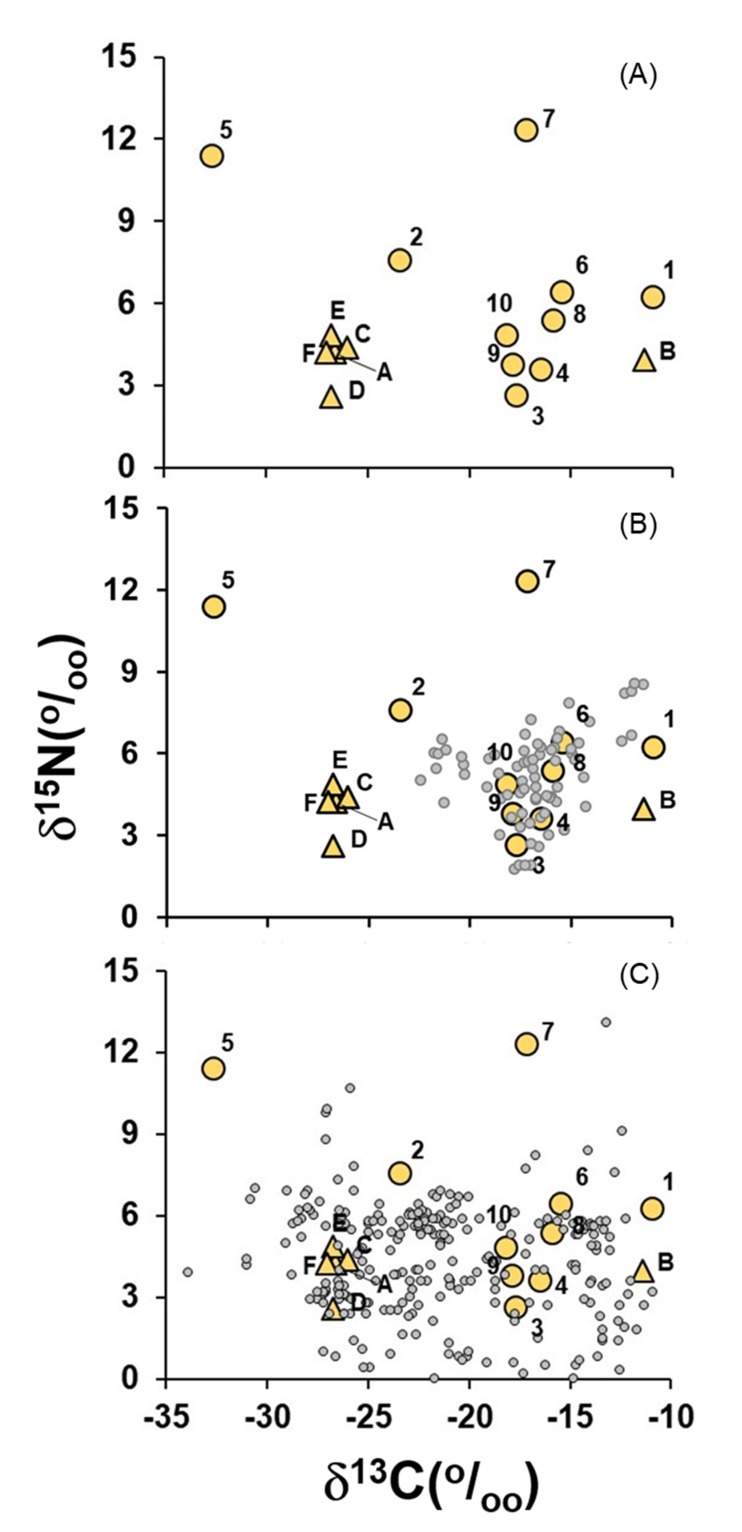
Biplot of δ^13^C vs. δ^15^N of: (**a**) plants (yellow triangle), and animal protein (yellow circle). A: C_3_-cereal; B: C_4_-cereal; C: fruit; D: pulse; E: tuber; F: vegetable. 1: beef; 2: bushmeat; 3: poultry; 4: egg; 5: freshwater fish; 6: lamb; 7: marine fish; 8: milk; 9: pork; 10: farmed fish (tilapia). (**b**) plants (yellow triangle), animal protein (yellow circle), processed food of animal origin (small grey circle). (**c**) plants (yellow triangle), animal protein (yellow circle), processed food of plant origin (small grey circle). In order to facilitate visualization, only processed foods with δ^15^N > 0‰ were included in (**b**,**c**).

**Figure 5 molecules-25-01457-f005:**
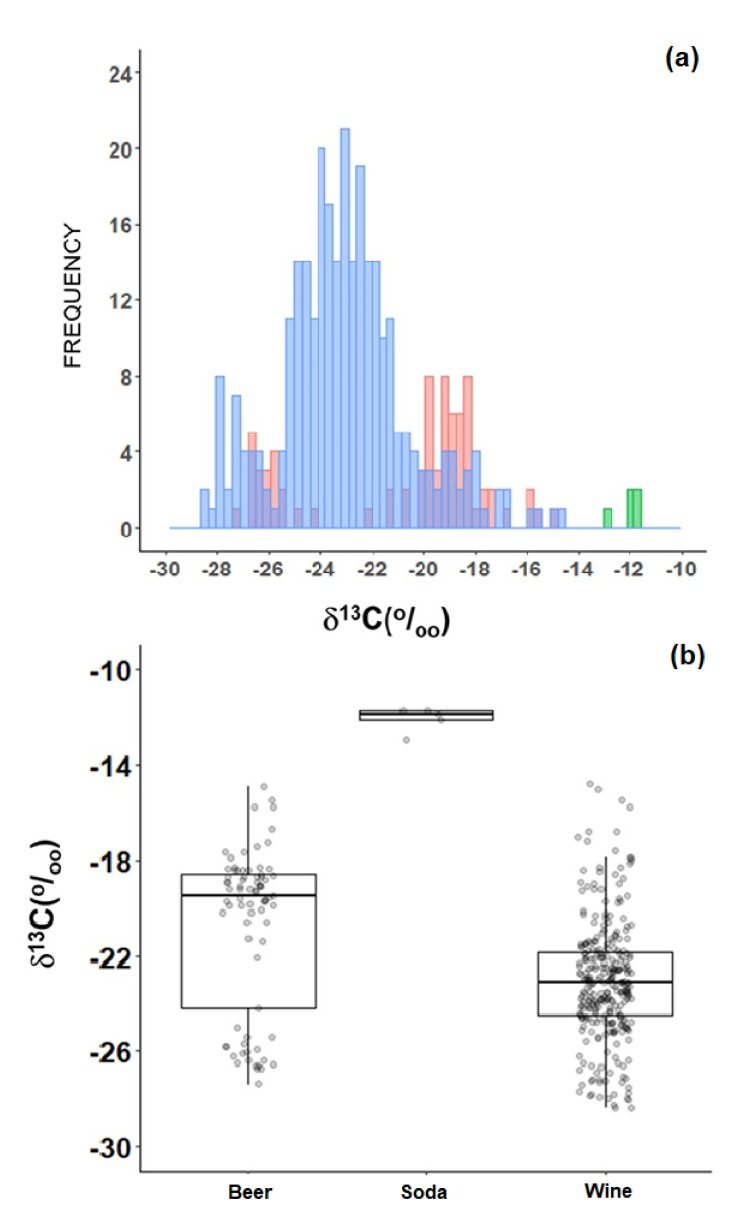
Frequency histogram (**a**) box-whisker (**b**) of δ^13^C of alcoholic beverages in Brazil. The blue bars represent wine samples, the green bars represent *cachaça* samples, and the red bars represent beer samples.

**Figure 6 molecules-25-01457-f006:**
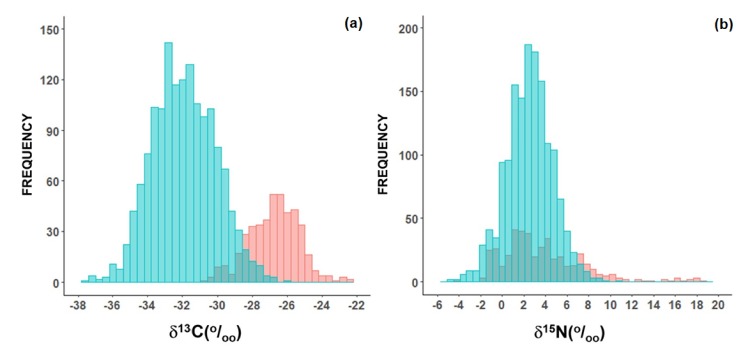
Frequency histogram of δ^13^C (**a**) and δ^15^N (**b**) comparing native tree leaves of the Atlantic Forest (blue bars) with domesticated plants used as food in Brazil (red bars).

**Table 1 molecules-25-01457-t001:** Descriptive statistics* of δ^13^C and δ^15^N of plant-based processed foods.

**δ^13^C (‰).**	**Mean**	**SD**	**Median**	**IQR**	**Min**	**Max**	***n***
Cake	−20.4	1.6	−20.2	2.4	−22.8	−18.2	11
Chocolate-bitter	−26.5	2.2	−26.1	2.0	−31.0	−24.3	19
Chocolate-drink	−14.4	1.1	−14.0	0.9	−17.2	−13.0	18
Chocolate-milk	−22.0	0.9	−22.2	1.3	−23.4	−20.1	28
Chocolate-powder	−18.8	2.3	−19.8	2.1	−20.8	−14.8	6
Chocolate-white	−21.9	0.8	−21.7	0.8	−23.8	−20.9	10
Cocoa powder	−27.8	2.3	−28.3	0.7	−29.1	−20.5	12
Cookie	−23.0	0.7	−23.1	0.8	−23.7	−21.4	12
Fat	−28.9	2.0	−30.1	2.6	−30.5	−25.4	8
Flour	−26.1	0.8	−26.0	0.9	−27.5	−23.9	20
Jam	−15.5	4.7	−13.5	2.8	−26.5	−11.6	33
Juice-powder	−13.0	1.1	−12.7	1.5	−14.7	−11.8	10
Noodles	−26.9	0.4	−27.1	0.8	−27.3	−26.2	9
Oil	−29.5	1.3	−30.0	1.5	−30.8	−27.4	7
Others	−17.7	4.5	−17.9	7.6	−24.2	−10.9	19
Pasta	−25.7	0.8	−26.2	1.3	−26.4	−24.6	5
Pudding	−12.8	2.3	−11.4	3.2	−16.1	−11.3	6
Sauce	−20.3	5.6	−20.0	9.9	−29.8	−12.3	27
Seasoning	−18.0	2.8	−19.5	3.3	−20.3	−13.7	5
Snack	−22.5	4.8	−21.0	8.8	−27.9	−17.0	9
Soda	−12.1	0.5	−11.9	0.4	−13.0	−11.7	5
Soup powder	−21.8	2.1	−22.6	1.0	−23.3	−17.7	6
Stock	−23.8	2.5	−24.6	2.4	−25.7	−21.0	3
Sugar	−12.2	0.3	−12.2	0.4	−13.2	−11.8	15
**δ** **^15^** **N (‰)**	**Mean**	**SD**	**Median**	**IQR**	**Min**	**Max**	***n***
Cake	3.7	0.3	3.6	0.1	3.3	4.3	11
Chocolate-bitter	5.9	0.7	5.8	0.7	4.4	7.3	19
Chocolate-drink	5.2	0.6	5.4	0.9	3.6	6.0	18
Chocolate-milk	5.8	0.4	5.8	0.5	5.1	6.8	28
Chocolate-powder	5.8	0.8	5.9	0.3	4.3	6.7	6
Chocolate-white	6.0	0.6	5.9	0.4	4.9	6.8	10
Cocoa powder	5.9	0.9	6.1	1.2	3.8	6.9	12
Cookie	3.7	0.9	3.8	1.1	2.4	5.1	12
Fat	2.5	2.4	3.8	2.1	−0.2	4.0	8
Flour	5.2	3.4	3.6	5.5	0.9	10.7	20
Jam	4.5	2.0	4.8	2.8	0.3	8.4	33
Juice-powder	−0.4	2.5	0.5	1.1	−5.8	1.8	10
Noodles	3.2	0.2	3.2	0.2	2.9	3.5	9
Oil	2.9	5.1	2.9	3.6	−0.7	6.5	7
Others	2.7	2.3	2.5	2.6	−0.8	7.7	19
Pasta	3.3	0.6	3.4	0.7	2.4	4.0	5
Pudding	5.9	0.1	5.9	0.1	5.8	5.9	6
Sauce	2.2	2.1	1.5	3.6	−1.4	5.8	27
Seasoning	−2.7	2.4	−3.1	3.2	−5.1	0.7	5
Snack	2.9	0.5	2.9	0.4	2.1	3.7	9
Soda	-^a^	-	-	-	-	-	-
Soup powder	2.0	1.3	2.0	0.8	0.0	4.0	6
Stock	−3.1	3.7	−3.7	3.7	−6.4	0.9	3
Sugar	-	-	-	-	-	-	-

* SD: standard deviation; IQR: inter quartile range; Min: minimum value; Max: maximum value. ^a^ Not determined.

**Table 2 molecules-25-01457-t002:** Descriptive statistics* of δ^13^C and δ^15^N of animal-based processed foods.

**δ^13^C (‰)**	**Mean**	**SD**	**Median**	**IQR**	**Min**	**Max**	***n***
Cheese	−16.0	2.0	−15.7	0.9	−22.4	−14.1	14
Cream cheese	−18.6	1.2	−18.6	1.5	−20.3	−17.4	5
Dehydrated stock cube	−19.5	5.2	−21.4	9.0	−25.8	−12.3	10
Ice cream	−21.0	1.1	−21.4	1.1	−21.8	−19.1	6
Jello	−12.0	0.4	−12.0	0.5	−12.5	−11.5	6
Lard	−19.9	2.8	−19.8	4.7	−23.4	−16.6	8
Processed meat	−16.8	0.9	−16.9	1.0	−18.5	−14.6	25
Sauce	−19.0	5.2	−19.6	8.1	−27.3	−13.5	11
Seasoning	−18.2	3.7	−17.1	5.8	−23.5	−14.8	6
Soup powder	−20.1	2.0	−20.5	3.2	−22.6	−16.6	12
Yogurt	−17.5	2.0	−17.3	1.7	−21.4	−14.3	13
**δ^15^N (‰)**	**Mean**	**SD**	**Median**	**IQR**	**Min**	**Max**	***n***
Cheese	5.9	1.2	5.9	1.8	4.2	7.9	14
Cream cheese	5.4	0.4	5.5	0.3	4.8	5.7	5
Dehydrated stock cube	−4.3	1.2	−4.5	0.9	−6.0	−2.0	10
Ice cream	5.9	0.2	6.0	0.2	5.5	6.1	6
Jello	7.8	0.9	8.3	1.4	6.5	8.6	6
Lard	4.3	1.1	4.0	1.0	3.3	5.9	8
Processed meat	4.2	1.7	3.8	2.8	1.8	6.7	25
Sauce	2.5	2.7	2.3	3.0	−1.8	6.9	11
Seasoning	−2.3	2.2	−3.1	3.4	−4.5	0.6	6
Soup powder	1.5	1.8	1.6	2.6	−2.3	4.4	12
Yogurt	5.1	0.7	5.1	1.2	4.1	6.5	13

* SD: standard deviation; IQR: inter quartile range; Min: minimum value; Max: maximum value.

**Table 3 molecules-25-01457-t003:** Comparison of average δ^15^N (‰) of domesticated plants reported this study and those in the meta-analysis of Huelsemann et al. (2015) [41]. Plants were grouped following the FAO classification. The values represent mean ± standard deviation.

FAO Group	This Study		Huelsemann et al. (2015) [41]	
	δ^15^N (‰)	*n*	δ^15^N (‰)	*n*
Cereal C_3_	3.7 ± 2.8	67	3.2 ± 1.3	>128
Cereal C_4_	4.5 ± 2.6	47	3.3 ± 1.9	57
Fruit	4.3 ± 3.0	81	4.6 ± 2.4	>891
Nuts	17.3 ± 0.8	10	2.7 ± 2.1	310
Pulse	2.4 ± 1.7	38	1.1 ± 1.8	109
Tuber	4.6 ± 2.8	45	3.1 ± 3.4	134
Vegetable *	4.2 ± 3.5	150	3.1 ± 3.0	>990

* Amended with mineral nitrogen fertilizer.

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
