# Peer review of "Carbon and Nitrogen Isotope Ratios of Food and Beverage in Brazil"

_molecules, 2020, doi:10.3390/molecules25061457_

Round 1

Reviewer 2 Report

Manuscript Number: molecules-717813

Title: Carbon and nitrogen isotope ratios of food and beverage in Brazil

It has been a pleasure reading through this contribution. By obtaining 1,678 food and beverage samples and analyzing for d13C and d15N isotopic composition, Martinelli et al. were able to build a relatively large database that could serve as a future reference for food and beverage provenance studies, with implications for regulatory applications in Brazil. Importantly, Martinelli et al. demonstrated that the d13C of Brazilian food is [one of] the highest in the world, a finding that is both well-grounded in the literature and potentially useful in making stable isotope forensics as a technique even more useful. I have no doubt that the broader community will stand to benefit from this contribution, and the underlying datasets, which may prove useful for other purposes. The community should also commend Martinelli et al. for making these datasets publicly available (upon publication). It is with excitement that I recommend immediate publication of this contribution as soon as the following points are addressed:

On the lack of a platform (e.g. hypotheses) upon which to anchor a story

As it stands, the manuscript reads more like a technical data note than a regular article-type contribution fitting for an MDPI journal. I suggest that the authors consider framing their potentially useful and impactful contribution against some hypotheses to test (statistically speaking). In my view, this would elevate the status of the paper from what now reads like a technical data note contribution to one that resembles an original, primary research paper. For example, given the prevailing understanding that C4 distribution is associated strongly with latitude (hence, climate), wouldn’t it be more interesting if the authors tested a null hypothesis (just as an example) to this effect and apply a decision-rule (i.e. statistical tests) to accept or reject such a hypothesis? I would presume that this is what partially motivated the research. But in its present form, the climatic control on C4 distribution is simply used as a confirmation to an otherwise already established fact (L353-360). Philosophically, framing a testable hypothesis could/would serve the readers well, especially in our pursuit to educate the future scientists about hypothesis testing in research.

Of course, this is not to say that exploratory research, which is self-evident in the current form of this contribution, is less useful than hypothesis testing (aka the scientific method) (see Pfister and Kirchner 2017, Water Resources Research). If the editor is okay with that then I will yield this comment to his/her better judgement. That said, it is my position that it would improve the value of the narrative of this contribution if such a suggestion was implemented. To extend this suggestion a little bit further, the authors may want to report tests for significance in respective plots, for example, Figure 2, Figure 3, and Figure 4b. In stable isotopes where the values of sample types typically span wide and overlapping ranges, it would be instructive to see whether any differences exist between samples (i.e. statistically).   

Moreover, the authors can employ resampling techniques (e.g. bootstrapping) to obtain much better and robust estimates of summary statistics, particularly for sample types with small sample sizes, e.g. see soda in Figure 4b.

On trophic isotopic discrimination

d13C predominantly receives attention in the paper, and that is understandable. However, the concept of trophic isotopic discrimination (L92) was never really explored in the results by way of an analysis other than a line to this effect in L331. I would suggest plotting a graph that demonstrates or supports L331 and the concept regarding trophic isotopic discrimination. This would be instructive and useful to see.

Minor comments

The manuscript could use some proofreading for some grammatical and minor spelling checks. Some of what I was able to identify follow:

L78: “as opposed to” instead of “opposed to”

L98-99: “a long”, not “along”

L110: delete “is” before “a municipality”

L123: change “mix” to “mixed”

L141: The factor 1000 in the equation is extraneous and unnecessary. Please remove (see Coplen 2011, RCMS)

L198: delete “a” before “C3-like”

L205: “are”, not “re”

L246: “were”, not “where”

L250: pi/pa: Please define and describe at first mention prior to any abbreviations. This was not mentioned before this line

Table 3: consider adding sample sizes of respective rows in both studies

L313: “that exist”

L320: “the production of which”, not “which its production is…”

L329: among all

L335: Did you mean “feed-free”?

L358: typo: patties

L364, L366: What is the journal rule on citations based on the web? Is it okay to just write the URLs in-text? Also, the font sizes of the URLs seem different than the text

L372: second largest producer

L373: fifth largest producer

L390: the need for

L391: increased sampling
